## PERSPECTIVE

# Accessory proteins of the RAS-MAPK pathway: moving from the side line to the front line

Silke Pudewell[1], Christoph Wittich[1], Neda S. Kazemein Jasemi[1], Farhad Bazgir[1] &
Mohammad R. Ahmadian [1✉]

Health and disease are directly related to the RTK-RAS-MAPK signalling cascade. After more than three decades of intensive research, understanding its spatiotemporal features is afflicted with major conceptual shortcomings. Here we consider how the compilation of a vast array of accessory proteins may resolve some parts of the puzzles in this field, as they safeguard the strength, efficiency and specificity of signal transduction. Targeting such modulators, rather than the constituent components of the RTK-RAS-MAPK signalling cascade may attenuate rather than inhibit disease-relevant signalling pathways.

Nature has evolved sophisticated, cell type-specific mechanisms to sense, amplify and integrate diverse external signals, and ultimately generate the appropriate cellular response. Signals are processed by evolutionarily conserved signalling cassettes that comprise specific constituent components acting as receptors, mediators, effectors and regulatory proteins. Activated receptor tyrosine kinases (RTKs), for instance, link the RAS activator SOS1 to RAS paralogs, e.g., the proto-oncogene KRAS4B, which in turn regulate various signalling pathways, including the mitogen-activated protein kinase (MAPK) pathway[1]. This pathway contains a three-tiered kinase cascade comprising the serine/threonine kinases ARAF/BRAF/CRAF, the dual specificity kinase MEK1/2 and the serine/threonine kinases ERK1/2[1,2]. The RTK-RAS-MAPK axis is a highly conserved, intracellular signalling pathway that has an essential role throughout mammalian development, from embryogenesis to tissue-specific cellular homoeostasis in the adult[3]. Dysregulation of components or regulators of this cascade is frequently associated with tumour growth and a distinct subset of developmental disorders called the RAS-MAPK syndromes or RASopathies[4–6]. This signalling cascade has rapidly taken centre stage in cancer and RASopathy therapies (see below).

However, the strength, efficiency, specificity and accuracy of signal transduction are controlled by mechanisms that increase the connectivity of the signalling molecules and thus increase their local concentration and reduce their dimensionality. This state can be achieved by liquid–liquid phase separation (LLPS), a mechanism in which two separate liquid phases with different protein compositions emerge from one mixed solution[7]. A large number of proteins, hereafter, collectively designated as the 'accessory proteins', fulfil the requirements to drive LLPS and have been reported to act as adaptor, anchoring, docking or scaffold proteins. Accessory proteins link constituent components of individual signal transduction pathways by forming physical complexes. What the functions of the accessory proteins are, why are they crucial for signal transduction, and whether they represent better therapeutic targets for different human diseases are questions that will be addressed in this article in the context of the RTK-RAS-MAPK signalling pathway.

**Structural and functional variety of accessory proteins**. Rapidly emerging reports on signalling networks support the idea that various signalling molecules operate together in functional protein complexes. For example, activated protein nanoclusters in specialised membrane

[1] Institute of Biochemistry and Molecular Biology II, Medical Faculty of the Heinrich-Heine University, Düsseldorf, Germany. ✉email: reza.ahmadian@hhu.de

microdomains selectively connect with and subsequently activate cytosolic signalling components or complexes[8,9]. RAS nanoclusters form and locally increase the concentration of RAS paralogs in membrane microdomains[10].

Membrane-resident signalling proteins, such as transmembrane (TM) and membrane-associated proteins, are predominantly trafficked to the plasma membrane via the secretory pathways[11]. But how are the cytosolic proteins trafficked to their cognate membrane nanoclusters? Mounting evidence has emerged recently that a large number of membraneless compartments (also called non-membrane-bound organelles or biomacromolecular condensates) are assembled via LLPS[12]. The formation of cytosolic signalling condensates is based on two processes. First multivalent molecules undergo phase separation, whereas in a second step other proteins are able to diffuse into the phase without considerably contributing to the stability of the phase. This process can increase local concentrations of molecules by several folds. One example is the enrichment of kinases in membrane-associated liquid droplets around T-cell receptors while phosphatases are excluded[13].

An essential group of proteins that are themselves not constituent components of signal transduction but allow assembly and spatiotemporal organisation of a signalling cascade are accessory proteins. These proteins have the features to interact with and assemble other biomolecules, ranging from lipids, over proteins to nucleic acids. They mostly lack enzymatic activity but are equipped with different types of protein–protein interaction domains, motifs and intrinsically disordered regions (IDRs). Thus, accessory proteins dictate the local formation of macromolecular protein complexes through modular multivalent interactions, and thereby organise and facilitate signal transduction.

Accessory proteins bind and connect at least two constituent components to orchestrate their spatiotemporal localisation and enhance their assembly by reducing the dimensionality of interactions and/or increasing local concentrations of interacting proteins[14–16]. They can be categorised in four distinct groups based on their structure and mode of action: (1) scaffold proteins are cytosolic multidomain proteins that bind two or more distinct components to organise them in a functional unit and modulate their function. (2) Adaptor proteins link two partners usually via SH2 and/or SH3 domains and may also regulate their specific downstream signalling pathways. (3) Anchoring proteins bind to the membrane and other proteins, which are usually protein kinases, and therefore, bring them to their site of action. (4) Docking proteins assemble signalling complexes by binding to effectors and RTKs or G-proteins at the membrane.

**Accessory proteins of the RTK-RAS-MAPK pathway**. New discoveries and concepts regarding the receptor-driven RAS-MAPK signal transduction have emerged during the last three decades: novel pathway components, structure elucidation, biophysical principles, biomimetic strategies and clinical drug candidates. By focusing particularly on the signalling process itself, the emphasis of this article is on the implementation of the accessory proteins, which bind molecular components and orchestrate their assembly and eventually activity in a context-dependent manner. We believe that the spatial arrangements of such biophysical features over time determine specificity, efficiency, fidelity of signal transduction and safeguard against any deleterious effects.

A multitude of accessory proteins, which largely vary in size and domain architecture (Fig. 1), are involved in orchestrating RTK-RAS-MAPK signal transduction. The high variability of scaffold proteins is—due to their high interaction specificity—comprehensible. Certain domains or repeats frequently exist in individual proteins, for example, LDs (repeated leucine-rich sequence) in Paxillin, WDs (WD-repeat) in MORG1, RRMs (RNA recognition motif) in nucleolin and LIMs in FHL1/2. Furthermore, IDRs are found in several proteins, which may fold upon interaction with their binding partner. IDRs are also involved in oligomerization for example in galectin-3[17]. Anchoring proteins contain membrane-binding domains, such as the PH domain in CNK1 and GAB1/2, and TM segment, e.g., in LAT, NTAL and SEF1. PAQR10/11 contain 7 TM segments and anchor RAS to the Golgi apparatus via their N-terminal cytoplasmic tail[18]. The PHB domain of FLOT1 has been reported to be a membrane association domain as it is post-translationally modified by palmitoylation[19]. This leads to FLOT1 association with lipid rafts of phagosomes and the plasma membrane. Docking proteins frequently possess both PH domains, which increase their residence time at the membrane, and PTB domains, which enable them to interact specifically with activated RTKs. Adaptor proteins are specialised in linking activated RTKs via SH2 domains with their downstream signalling molecules, in most cases, via SH3 domains.

**Linking TM receptors to RAS**. GRB2 links activated RTKs or anchoring proteins, such as LAT, with SOS1/2 to activate RAS paralogs (Fig. 2a)[20]. The adaptor protein function of GRB2 is accomplished by a central SH2 domain that binds to the tyrosine-phosphorylated RTK and two flanking SH3 domains, which bind to the C-terminal proline-rich domain of SOS1 and translocate it to the plasma membrane[21,22]. Activated SOS1, in turn, stimulates, as a RASGEF, the GDP/GTP exchange of RAS paralogs and thereby activates amongst others the MAPK cascade[23].

Furthermore, direct GRB2 association with activated RTKs leads to the recruitment of GAB1 and CBL. GAB1 provides a docking platform for several signalling molecules, e.g., SHP2, PLCγ and PI3K, thereby cross-linking different signalling pathways[24]. CBL was originally described to act as an adaptor protein as it contains several domains and motifs for protein–protein interactions (Fig. 1). Later, it was identified as a RING-dependent E3-ubiquitin-protein-ligase that transfers the ubiquitin to RTKs for endocytic internalisation, and recycling or degradation[25]. It also regulates signalling processes of the non-RTKs SYK, ZAP70 and SRC[26]. CBL constitutively interacts with GRB2, mediating hematopoietic cell proliferation[27], and T-cell and B-cell receptor and cytokine receptor signalling via interaction with CRKL SH2 domain[28]. As CBL and SOS1 bind to the same region of GRB2, the overexpression of CBL inhibits complex formation between SOS1 and GRB2 underlining the fine-tuning mechanism of accessory proteins by binding other pathway modulators[29].

Engagement of GRB2 is versatile and leads to different outcomes depending on the cellular context. GRB2 can bind indirectly to RTKs via interaction with the tyrosine-phosphorylated adaptor proteins SHC and FRS2. SHC links activated TRKA receptors to GRB2 in PC12nnr5 cells[21,22,30], which can recruit SOS to the PM and control the extent of RAS activation[23]. Upon activation of the B-cell antigen receptor (BCR) in B-lymphocytes, the tyrosine kinase SYK phosphorylates SHC which leads to translocation of GRB2-SOS1 and activation of membrane-associated RAS signalling[31]. The SHC–GRB2 complex, downstream of cytokine receptors, also activates the PI3K pathway to control cell survival and/or proliferation[32]. A similar mechanism of GRB2-SOS-RAS activation is operated via FRS2, which acts downstream of TRKA in neurons[21], and FGFR in embryonic stem cells[33,34]. FRS2 has multiple tyrosine phosphorylation sites to activate, in response to a wide range of agonists, PI3K and RAS-MAPK pathways in various cell types via binding

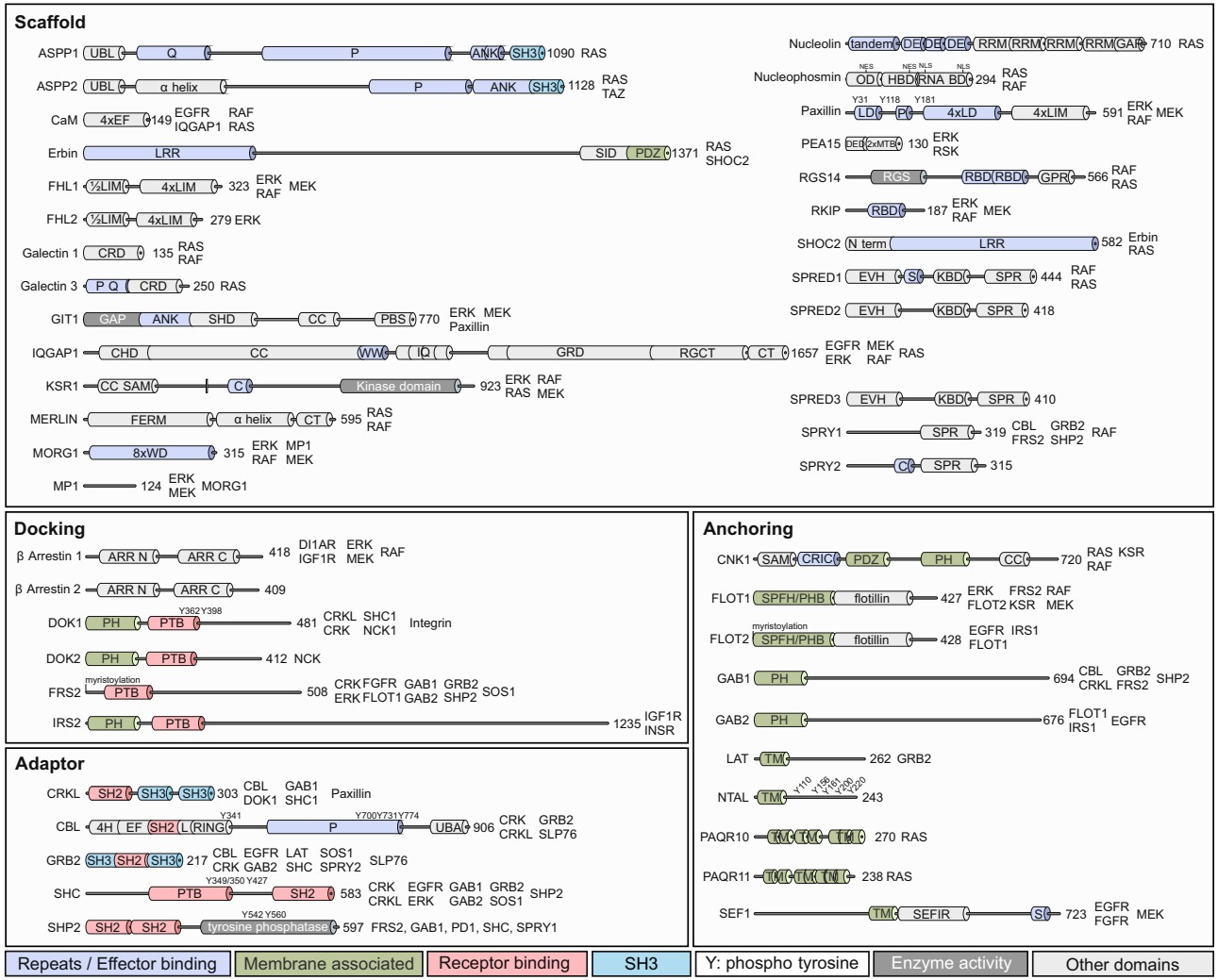

**Fig. 1 Domain organisation and crucial interactions of RTK-RAS-MAPK accessory proteins.** Schematic representation of relevant domains in scaffold, docking, adaptor and anchoring proteins. Direct binding partners, which are part of the RTK-RAS-MAPK pathway, are mentioned next to the amino-acid numbers at the right side of the proteins. Please check the list of abbreviations in the supplemental table for more details. Abbreviations are summarised in Supplementary Table 1.

to GRB2 and SHP2, respectively[35–39]. The binding of the ubiquitous protein tyrosine phosphatase SHP2 to GRB2, induces recruitment by the FRS2-SHP2 complex, which controls retinal precursor proliferation and lens development[40].

**Modulating the RAS cycle.** The RAS cycle between an inactive, GDP-bound state and an active, GTP-bound state is strictly controlled by multidomain regulatory proteins[41–44]. Unlike the well-understood cellular process of RAS activation by RASGEFs, such as SOS1 little is known about the recruitment and activation of RASGAPs. The first evidence has emerged that the RASGAPs neurofibromin and p120 are recruited to the plasma membrane and RAS•GTP by two distinct scaffold proteins, SPRED1 and merlin (Fig. 2b). The EVH domain of SPRED1, a member of the sprouty family, binds the GAP domain of neurofibromin without interfering with its GAP function[45,46]. SPRED1 appears to directly contact BRAF and thus to interfere with KRAS signalling[47]. Merlin, a member of the ERM family, directly binds to, on the one hand, p120 and RAS (probably KRAS4B), a mechanism that potentiates RAS inactivation in Schwann cells, and on the other hand, CRAF and blocks its interaction with RAS[48,49]. p120 modulates many regulators and signalling

proteins via its N-terminal protein interaction domains, apparently independent of its GAP function[50,51].

**RAS-RAF connection.** Lipidation and clustering of the RAS paralogs are critical steps for a tight control of signal transduction through the MAPK pathway. This process connects two distinct macromolecular clusters, plasma membrane-associated RAS-containing clusters[9] and cytosolic RAF/MEK/ERK-containing clusters[52].

The scaffold proteins galectin 1 and 3 are carbohydrate-binding proteins that are involved in many physiological functions. While galectin 1 homodimer binds to HRAS-RAF complex and stabilises HRAS•GTP at the plasma membrane[10,53], galectin 3 selectively binds and clusters KRAS4B•GTP (Fig. 2c)[54]. The nucleolar phosphoproteins nucleophosmin and nucleolin shuttle between nucleus and PM and are different types of RAS scaffold proteins, which have been reported to stabilise KRAS4B levels in a nucleotide-independent manner at the plasma membrane. Nucleophosmin also increases the KRAS4B•GTP clusters and enhances MAPK signal transduction[55].

Another type of clustering is performed by the scaffold protein SHOC2 (also known as SUR8), which connects activated RAS with the RAF kinases (Fig. 2d). SHOC2 is an integral element of a heterotrimeric holoenzyme complex with PP1 and MRAS, which

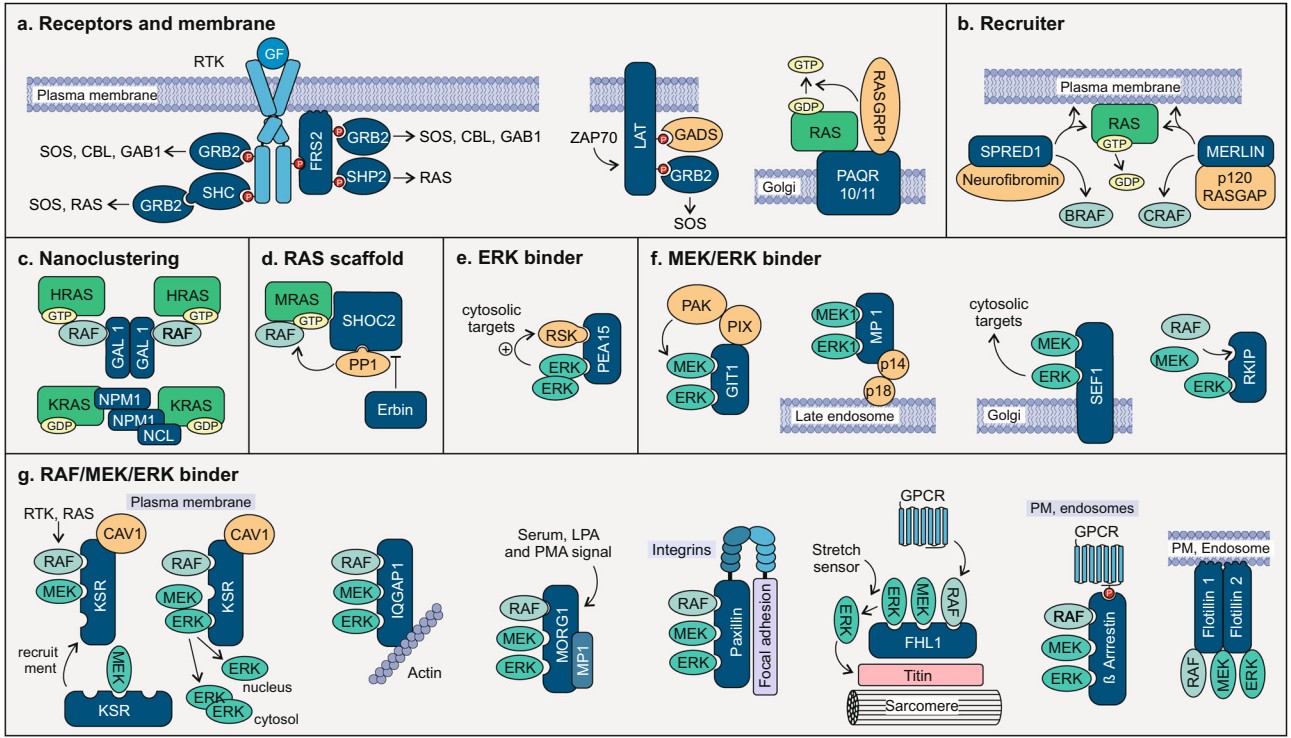

**Fig. 2 Involvement of RTK-RAS-MAPK accessory proteins in signal transduction.** Accessory proteins are involved in every step of the RTK-RAS-MAPK pathway and increase the connectivity between signalling components. Adaptor proteins and docking proteins interact with phosphorylated receptors, in contrast to anchoring proteins, which are directly associated with the membrane (**a**). Recruiter translocate binding partner towards the site of action, e.g., a GTPase activating protein to activated RAS at the plasma membrane (**b**). Scaffold proteins that induce nanoclustering, increase the local concentration of their binding partner in lipid rafts (**c**). RAS scaffold proteins bind RAS and other components of the RTK-RAS-MAPK pathway (**d**), whereas ERK binder (**e**), MEK/ERK binder (**f**) and RAF/MEK/ERK binder (**g**) can connect one, two, or all three members of the MAPKs, bringing them close together, regulate their activity and determine their localisation. See text for more details.

dephosphorylates and releases RAF from its inhibited state[56,57], and subsequently activates the MAPK pathway[58]. The scaffold protein Erbin interferes with this process[59]. It binds and sequesters SHOC2 from its RAS/RAF complex, and inhibits ERK activation[60]. Erbin is a large scaffold protein (Fig. 1). As such, it links different pathways by binding, besides SHOC2, also various other accessory proteins, including GRB2[61], CBL[62], Merlin[63] and KSR1[2,64].

**RAF/MEK/ERK cascade.** RAF kinase translocation to the plasma membrane and activation by direct interaction with RAS•GTP is well described[2,65–67]. Activated BRAF/CRAF heterodimer phosphorylates MEK1/2, which in turn phosphorylates ERK1/2 at the TEY motif in the activation loop[68,69]. Activated ERK1/2 are ultimately recruited to their substrates in various subcellular compartments[70,71]. The assembly of macromolecular complexes of the MAPK components and their connection with RAS nanoclusters at the membrane, which constitutes the RAS–ERK axis, is arranged by homo- and heterodimerization of the members of this pathway[68]. To achieve signal diversity, specificity and fine-tuning, the spatiotemporal flux through the pathway is organised by various distinct accessory proteins, which bind either ERK, MEK/ERK, or RAF/MEK/ERK[1,70,72].

PEA15 modulates ERK activity towards its cytosolic substrates, including RSK2. It enhances ERK-dependent phosphorylation of RSK2 by binding both of them independently (Fig. 2e)[73]. PEA15 phosphorylation by PKC, AKT, or CaMKII inhibits this process. In addition, PEA15 steers subcellular localisation of ERK by facilitating its nucleocytoplasmic export[74].

The MEK/ERK accessory proteins are illustrated in Fig. 2f. GIT1 binds MEK1 and ERK1 in response to integrin, RTK and GPCR activation. Its activity is directly regulated by different downstream effectors, such as PIX/PAK complex[75]. MP1 binds and translocates MEK1 and ERK1 to late endosomes by associating with p14 and p18[76,77]. The anchoring protein SEF binds activated MEK on the Golgi apparatus, and subsequently binds ERK, leading to activation of ERK and finally its cytosolic substrates such as RSK2[78]. The latter phosphorylates SEF and induces its translocation to the plasma membrane, where it directly inhibits FGFRs, and enhances EGFR signalling instead[79]. RKIP acts as a competitive inhibitor of MEK phosphorylation. It binds ERK and mutually exclusively RAF or MEK, and thus, dissociates active RAF/MEK complexes[80]. The phosphorylation of RKIP by PKC results in the release of RAF1 and enables the activation of the MAPK pathway[81].

The scaffolding of RAF/MEK/ERK is dependent on several factors, including the tissue specificity, cellular localisation of the signalling complexes and the type of upstream signals (Fig. 2g). KSR1 is one of the best-studied scaffolds that binds to all three members of the RAF/MEK/ERK cascade[72]. KSR1 translocates, upon RTK-RAS activation, in a complex with MEK to CAV1-rich microdomains in the plasma membrane to bind activated RAF and modulate MEK and ERK activation. Feedback phosphorylation of KSR1 and BRAF by ERK promotes their dissociation and results in the release of KSR1/MEK from the plasma membrane[82]. In this way, MEK is sequestered from upstream signals and cannot itself regulate ERK activation.

The multidomain protein IQGAP1 scaffolds and activates the RAF/MEK/ERK kinases by directly associating with the EGF

receptor[83,84]. With over 100 binding partners, the localisation and effect of IQGAP1 interaction reach from actin cytoskeleton reorganisation in the context of neurite outgrowth, migration or vascular barrier integrity to insulin secretion via exocytosis or cell proliferation and differentiation via ERK signalling. The extensive interactions of IQGAP1 vary according to cell types and environmental conditions[85]. In contrast, MORG1, FHL1, paxillin and β-arrestin act EGF-independent (Fig. 2g). MORG1 exists in a complex with MP1 and facilitates ERK1/2 activation in response to LPA and PMA, and GPCR activation[86]. The focal adhesion protein paxillin modulates the activation of the RAF/MEK/ERK complex through the binding of other proteins, controlling the remodelling of the actin cytoskeleton[87]. FHL1 scaffolds RAF/MEK/ERK on the N2B domain of the giant protein titin at the sarcomere of the mammalian muscle cells[88]. β-arrestin stimulates ERK signalling in response to activation of GPCR or other receptors on the plasma membrane but also on endosomes. FLOT1/2 are membrane raft-associated proteins that form heterodimers. They are not only involved in the EGF receptor clustering and activation, but also directly bind CRAF, MEK and ERK enhancing their activity upon stimulation[89]. CNK1 physically interacts with RAF facilitating its activation by assisting RAF membrane localisation and oligomerization upon RAS activation[90], whereas being able to interact with RAS as well via the N-terminal regions[91].

**Accessory proteins as in human disease.** Even if dysregulated constituent components of the RTK-RAS-MAPK pathway are among the most intensively studied target structures for disease treatment, new emphasis should be laid on accessory proteins (Fig. 3). Their loss-of-function or gain-of-function mutations are mostly and frequently associated with the initiation and progression of human diseases and disorders. The hyperactivation of the RTK-RAS-MAPK pathway is a known cause of many diseases, like cancer and developmental disorders, including RASopathies.

**Cancer.** The upregulation of activating proteins or the downregulation of inhibiting proteins leads to gain-of-function of the RTK-RAS-MAPK pathway in almost all types of cancer (Fig. 3a). The expression of accessory proteins is tightly controlled and often dysregulated in tumours. Paxillin is a scaffold protein, which is involved in focal adhesion. A gain-of-function mutation in *Paxillin* has been found in 9% of all non-small cell lung cancers (NSCLC) (1)[92]. Furthermore, genomic amplification of *Paxillin* in lung cancer promotes tumour growth, invasion and migration[93]. SPRED1/2, negative modulators of RAS signalling, are downregulated in 84% of patients with hepatocellular carcinoma (2)[94]. The scaffold protein IQGAP1 promotes tumour formation,

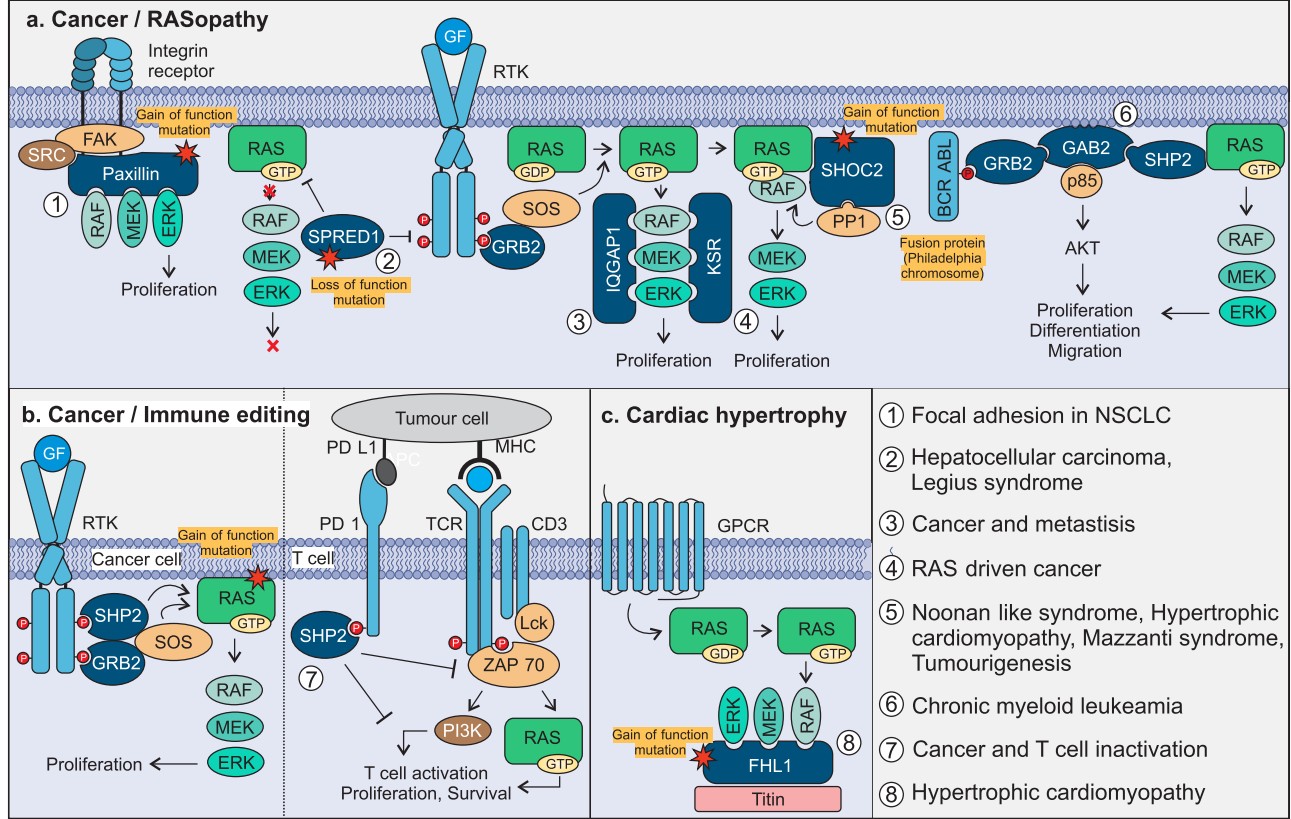

**Fig. 3 Involvement of accessory proteins in diseases.** The canonical RAS/MAPK pathway is tightly regulated by many proteins, attenuators and negative feedback mechanisms. Mutations in regulators like accessory proteins can lead to a dysregulated RAS/MAPK pathway and therefore to a variety of diseases as cancer and RASopathies (**a**). The genomic amplification of Paxillin is found in many NSCLC patients and activates the focal adhesion complex downstream of integrins (1). Loss-of-function mutations of SPRED1 activate the RAS-MAPK pathway and lead to Legius syndrome (germline) and hepatocellular carcinoma (somatic) (2). IQGAP1 mutations are often associated with tumour formation and metastasis (3), whereas KSR is a central player in KRAS-driven cancers, inducing proliferation and survival (4). Mild gain-of-function mutations of SHOC2 lead to Noonan-like syndrome with loose anagen hair or Mazzanti syndrome, other somatic mutations can lead to hypertrophic cardiomyopathy or tumourigenesis (5). The signalling of BCR-ABL-positive cells in chronic myeloid leukaemia is also dependent on GAB2 activation, cross-linking AKT and RAS pathway (6). The adaptor protein SHP2 is not only part of hyperactive RAS signalling in cancer cells, but is of major importance in the inactivation of T cells, inhibiting the TCR signal in response to ligand binding to PD-1 (**b**). FHL1 is involved in the development of cardiac hypertrophy, which is caused by a gain-of-function mutation, leading to increased ERK signalling (**c**).

transformation, invasion and metastasis in various cancer types (3)[95]. A study of a KSR$^{-/-}$ mouse model proves the resistance against RAS-dependent tumour formation[96], highlighting the pro-oncogenic function of KSR in RAS-driven cancers (4). SHOC2 mediates tumourigenesis and metastasis in different cancer types via tethering RAS and CRAF proteins in close proximity and thus promoting RAS-mediated CRAF activation[97,98]. Knockout models of SHOC2 in KRAS mutated lung adenocarcinoma in mice have revealed a significant reduction of tumour growth, as well as a prolonged survival, accentuating the scaffold protein as a potential therapeutic target (5)[99]. GAB2 has been implicated as a central modulator for oncogenic BCR-ABL signalling[100]. GAB2-deficient mice have exhibited resistance against cancer cell transformation of myeloid progenitors in the presence of BCR-ABL, which is found in 90% of patients with chronic myeloid leukemia (6)[100,101]. SHP2 is not only associated with a large number of cancers but plays a central role in PD-L1/PD-1 singling that inhibits the TCR-activated pathways, including RAS-MAPK, in T cells (Fig. 3b (7))[102]. This leads to an inactivation of the T cells, guarding the tumour cells against the immune system. Thus, SHP2 inhibitors have a dual role as a possible therapeutic target by reducing RAS signalling and inducing the body's immune response.

**RASopathies**. RASopathies or RAS-MAPK syndromes are defined as a group of developmental disorders that are caused by mild gain-of-function germline mutations in genes related to not only the constituent members of the RTK-RAS-MAPK pathway[103] but also various accessory proteins, including CBL, SHP2, SPRED1 and SHOC2 (Fig. 3a)[103].

Germline *CBL* mutations exhibit a wide phenotypic variability related to Noonan syndrome, which is characterised by a relatively high frequency of neurological features, predisposition to juvenile myelomonocytic leukaemia and low prevalence of cardiac defects, reduced growth and cryptorchidism[104]. The mutations are mainly located in the central region of CBL, which is known to abolish the ubiquitination of RTKs by impairing CBLs E3 ligase activity[104]. Legius syndrome-associated mutations in *SPRED1*, mostly result in loss-of-function of the scaffold protein, and gain-of-function of the RAS-MAPK pathway[105,106]. In contrast, mutations in genes encoding SHP2 and SHOC2 lead to a gain-of-function and contribute to MAPK signalling upregulation that causes diverse developmental phenotypes[56,59,107]. A recurrent activating mutation at the very N-terminus of SHOC2 (Ser-2 to Gly) leads to N-myristoylation of SHOC2, confers continuous membrane association and consequently causes Mazzanti syndrome, a RASopathy characterised by features resembling Noonan syndrome[107,108]. Another RASopathy-causing *SHOC2* mutation (Gln-269 to His and His-270 to Tyr) has been recently identified to be associated with prenatal-onset hypertrophic cardiomyopathy[107]. This mutation changes the relative orientation of the two leucine-rich repeat domains of SHOC2 and enhances its binding to MRAS and PPP1CB, two other RASopathy genes[109], and thus, increased signalling through the MAPK cascade[107].

**Other diseases**. Moyamoya angiopathy is characterised by progressive stenosis of the terminal portion of the internal carotid arteries and the development of a network of abnormal collateral vessels. This is a rare condition that can be caused by de novo *CBL* mutations even in the absence of obvious signs of RASopathy[110]. Evidence linking CNK1 dysfunction to autosomal recessive intellectual disability in patients emphasises the importance of this anchoring protein in the orchestration of the RTK-RAS-MAPK signalling in brain development and

cognition[111]. The scaffold proteins FHL1/2 link RAS-MAPK signalling to the sarcomere and is a critical component of the hypertrophy signalling in cardiac cells (Fig. 3c)[88]. FHL1/2 mutations are associated with cardiac diseases[112]. FLOT1 has been implicated in the development of Alzheimer and type 2 diabetes and could be a promising proteomic biomarker[113,114].

**Accessory proteins as therapeutic targets**. Direct targeting of constituent members of the RTK-RAS-MAPK axis in the context of disease treatment, such as cancer, is a big challenge. Therapies for KRAS mutated cancers remain a major clinical need, despite allele-specific inhibitors that trap and inactivate mutant KRAS (G12C)[115,116]. Three decades of research led to significant advances in tumour treatment[117]. However, the side-effects can still be severe and more-specific treatments could ease patient suffering. Unfortunately, many of the expectations for RAS pathway-targeted drugs have not been fulfilled. High toxicity and resistance acquisition have hampered many of the drugs developed to date[117,118]. An alternative therapeutic strategy to treat KRAS mutant cancers aims at protein degradation via proteolysis targeting chimeras (PROTACs)[119]. The ablation of CRAF in advanced tumours driven by KRAS oncogene leads to significant tumour regression with no detectable appearance of resistance mechanisms and limited toxicities[120]. In this context, a recent study has reported first progress to develop degrader molecules that target KRAS oncogene in NSCLC[121].

Emerging evidence suggests that constituent signalling proteins assemble into macromolecular complexes and co-operate in clusters at specific sites of the cell. Therefore, it is important to note that the stoichiometric imbalance of each subunit of a complex—either by gene overexpression on the one side, and depletion, knockout or targeted protein degradation on the other—perturbs the equilibrium, and interferes at some level with the function of the protein or its complex[122]. With accessory proteins being of immense relevance for the whole signalling machinery and operating particularly from the side line, we propose that functional interference with a defined site of accessory proteins may attenuate rather than inhibit the signalling of hyperactivated RTK-RAS-MAPK axis.

The knockout or knockdown of accessory proteins in cell-based or animal models could already show the importance of these modulators in cancer signalling. The scaffold protein SHOC2 has an important role in embryogenesis, therefore, loss-of-SHOC2 is embryonically lethal. In contrast, the systemic knockout in adult mice as well as in human cell lines is quite well tolerated and leads to growth inhibition of RAS-mutated NSCLC cell lines[99]. Furthermore, the depletion of SHOC2 leads to a sensitisation towards MEK inhibitor treatment, by interfering with the feedback-loop of MEK inhibition via BRAF/CRAF dimerisation, which is SHOC2 dependent[99]. Therefore, dual targeting of SHOC2 and MEK appears as a promising treatment strategy in RAS-mutated cancers. Another approach deals with the scaffold protein GIT1. The knockdown of GIT1 in human osteosarcoma cells has shown in vivo and in vitro reduced tumour cell growth, invasion and angiogenesis, which could make GIT1 a potential target in gene therapy[123].

There is a number of approaches to target specific functions of accessory proteins (Table 1). The CNK1 inhibitor PHT-7.3 binds to its PH domain and prevents the colocalisation with prenylated KRAS4B on the plasma membrane[124]. PHT-7.3 successfully inhibits the growth of tumour cells induced by mutated but not wild type KRAS4B. The interference of GRB2 mRNA by liposome-incorporated nuclease-resistant antisense oligodeoxynucleotides in BCR-ABL fusion protein-positive cancer cells, leads to reduced tumour growth in Xenograft models[125]. It

**Table 1 Accessory proteins as attractive therapeutic targets.**

| Accessory protein | Disease | Drug | State of art | Comment | Ref. |
|---|---|---|---|---|---|
| CNK1 | Cancers with KRAS mutations | PHT-7.3 | Cell-based model | PHT-7.3 binds selectively to CNK1 PH domain, interferes its colocalisation with KRAS4B on the plasma membrane and diminishes RAF/MEK/ERK signalling | 124 |
| GRB2 | BCR-ABL-positive leukaemia | Anti-miDNA L-GRB2 | Xenograft model | L-GRB2 selectively targets GRB2 mRNA and inhibits its translation | 125 |
| IQGAP1 | Cancers with KRAS mutations | WW competitive peptide | Mouse model | WW competitive peptide antagonist of IQGAP1 interferes with its scaffolding ERK interaction; it is applied in combination with the BRAF inhibitor vemurafenib (PLX-4032) against KRAS4B oncogene | 126 |
| KSR | Cancers with KRAS mutations | APS-2-79 | Cell-based model | APS-2-79 binds and stabilises KSR in its inactive state, interferes with KSR/RAF heterodimerization and inhibit oncogenic KRAS4B signalling | 128,129 |
| SHP2 | Cancers | SHP099 | Xenograft model | SHP099 binds SHP2 as an allosteric inhibitor, stabilises its autoinhibited state and inhibit oncogenic Ras signalling | 130–133 |

interferes with the RAS/MAPK pathway and the cross-talk towards AKT pathway via GAB2. A WW-peptide of IQGAP1 binds ERK and competes with endogenous IQGAP1, which leads to attenuation of ERK activation[126]. This treatment together with the BRAF inhibitor vemurafenib (PLX-4032), was very successful in tumour mouse models[126]. It has later been shown that not the WW-domain but the IQ domain is necessary to bind ERK[127]. The effects on the tumour growth suppression may stem from the interference with another yet unknown binding partner of IQGAP1 as an integral element of its complex scaffolding function. Another interesting example of accessory proteins as a therapeutic target is the small molecule APS-2-79, which binds KSR in its inactive state and interferes with RAF binding and thus blocks MEK phosphorylation[128]. The cell-based experiments with APS-2-79 have shown not only reduced ERK activation and growth inhibition in combination with the MEK inhibitor trametinib, but also antagonising its resistance mechanism[129]. Besides active site inhibitors, an allosteric inhibitor of SHP2 SHP099 stabilises the autoinhibited state and interferes with the enzymatic activity as well as its adaptor protein function to bind, for example, the GRB2-SOS complex[130]. A combination of SHP099 with a MEK inhibitor has been shown to interfere with the feedback mechanism via SHP2 and to block the resistance initiation observed in KRAS4B-driven cancer therapy[130–132]. In addition, SHP2 inhibition by SHP099 has been shown to have a positive effect on anti-tumour immunity in colon cancer xenograft models, especially in a co-treatment with an anti-PD-1 antibody[133].

Given that the majority of accessory proteins are now emerging as attractive therapeutic targets, still a very small number of accessory inhibitors have been discovered yet.

## Concluding remarks and outlook
Accessory proteins tightly control signal transduction by fine-tuning spatiotemporal organisation of signalling components and maintaining specificity and function of the pathway on a cell type and even subcellular level. They operate from the side line, from which they specifically leverage their multivalent domains on the formation of macromolecular clusters, as highlighted in this article. Even though interest in accessory proteins has grown in the past few years, the possibilities to practically visualise them, track their pathway and experimentally and selectively affect their functions in human cells are keys to address questions about their cell type specificities, subcellular distribution and physical interactions in a context-dependent manner. To investigate the impact of an accessory protein in the context of RAS-MAPK signalling,

we suggest the following approach: (i) It is necessary to first determine a cell line that expresses the gene related to the accessory protein of interest using quantitative real-time PCR. (ii) It is crucial to investigate the accessory protein at the endogenous levels. The overexpression studies cause in spite of their experimental advantages various difficulties[122]. A prominent example is KSR overexpression that has been erroneously identified as a suppressor of RAS signalling. (iii) The major challenges faced and likely to be faced in near future are the difficult task of the direct use of antibodies post-purchase without careful validation[134]. It is of major importance to validate the antibody specificity by immunoblotting purified protein or protein fragments, and cell lysates overexpressing gene or gene fragments encoding the accessory protein. (iv) Cell fractionation and confocal imaging under-stimulated and non-stimulated conditions will prove if the proteins pre-assemble in complexes with their binding partners (as predicted for KSR-MEK) and where they are located within the cell; as we expect the accessory proteins to orchestrate the RTK-RAS-MAPK signalling in specific subcellular compartments (e.g., plasma membrane, early endosomes, lysosomes, Golgi or ER). (v) Gene knockout cell lines, generated by CRISPR/Cas9 technology, will allow measuring the impact of the accessory proteins as positive or negative modulators of the RAS-MAPK pathway, by determining the p-ERK/ERK ratio. Moreover, this approach will give an idea about possible feedback or compensation mechanisms of accessory proteins among each other. Thus, exploring these concepts in greater detail will provide the framework for future research that will fill existing gaps in our knowledge and expand our understanding of more effective therapies.

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

## Acknowledgements

We are grateful to our colleagues Ehsan Amin, Ion C. Cirstea, Oliver Krumbach, Niloufar Mosaddeghzadeh, Saeideh Nakhaei-Rad and K. Nouri, for stimulating discussion. This study was supported by the Research Committee of the Medical Faculty of the Heinrich Heine University Düsseldorf (grant number: 2020-70/9772617), the German Research Foundation (Deutsche Forschungsgemeinschaft or DFG; grant number: AH 92/8-1), the German Research Foundation (Deutsche Forschungsgemeinschaft or DFG) through the International Research Training Group 'Intra- and interorgan communication of the cardiovascular system' (grant number: IRTG 1902-p6), the European Network on Noonan Syndrome and Related Disorders (NSEuroNet; grant number: 01GM1621B); the German Federal Ministry of Education and Research (BMBF)—German Network of RASopathy Research (GeNeRARe; gant numbers: 01GM1902C).

## Author contributions

S.P., N.S.K.J., F.B. and C.W. systematically searched and read the literature using the PubMed database; C.W. generated Fig. 1, and S.P. generated Figures 2 and 3. All authors, including M.R.A. designed, wrote and approved the final version of the manuscript.

## Funding

## Competing interests

The authors declare no competing interests.
