## [Peer Review File · Communications Biology]

Reviewers' comments:

Reviewer #1 (Remarks to the Author):

Title; Review of Accessory proteins of the RAS-MAPK pathway: Moving from the sideline to the front line

Paper summarizes the field of liquid-liquid phase separation in respect to RAS-RAF-MEK-ERK pathway. Previously, authors did discover the Galectin interaction with Ras and have written couple of reviews.

Any insights and implications? What are the major claims of the paper? Are they novel and will they be of interest to others in the community and the wider field?

Manuscript does provide novel insights into signaling cascade activation and highlights importance of scaffolding function of proteins on forming liquid-liquid phase separation.

Issues? Is the work convincing, and if not, what further evidence would be required to strengthen the conclusions? On a more subjective note, do you feel that the paper will influence thinking in the field?

Authors made a convincing point on importance of accessory proteins on signal activation and control of localization signal within the cells and cellular compartments. Whether paper will influence the thinking in the field is heavily depends on upcoming research articles and advancements in the field. I believe that scaffold proteins are important but field is still trying to distinguish between allosteric activations versus increased proximity for activation as it was the case for KSR protein.

1. Authors explain and exemplified in depth the accessory proteins that impact on signaling cascade activation. Could authors elaborate potential of these accessory proteins as synthetic lethal pairs in cancers as a part of "accessory proteins as therapeutic targets" section?

2. Technological limitations for studying impact of each accessory protein in signaling cascade could be mentioned. While these proteins are critical, studying their functional nuances require better assays on pathway activation and spatiotemporal movement of individual proteins in the pathway.

Following the same comment,

A. How strength of each accessory proteins varies and how researchers could identify novel accessory proteins? How to emphasize the temporary interactions on signaling pathway and how does feedback loops fit into stability of the activation clusters?

B. Authors mentions array of proteins that functions as scaffold protein in RAS/RAF/MEK/ERK pathway as shown in figure 1. How many of these scaffold proteins needs to be present for enhanced signaling at a given time? Could cell compensate for inhibition of specific protein by increased expression of another scaffold protein?

Overall, manuscript decently summarizes the field. With minor edit, it could be accepted.

We would also be grateful if you could comment on the appropriateness and validity of any statistical analysis, as well the ability of a researcher to reproduce the work, given the level of detail provided.

NA

Reviewer #2 (Remarks to the Author):

This is a well-researched review article that focuses on the functions of accessory proteins that control specific nodes of the RTK-RAS-MAPK signalling pathway. Accessory proteins are explained in terms of their function and are divided into i) scaffold, ii) adaptor, iii) anchoring, and iv) docking proteins, followed by highlights of well-studied domains, and the role they play in interfacing and controlling pathway dynamics. The second half of the article focuses on how dysfunction of RAS-MAPK accessory proteins can lead to diseases such as cancer. The major claim of the article is that RTK-RAS-MAPK accessory proteins should be a major focus of experimental cancer therapeutics going forward. The article is persuasive and makes a strong case for targeting accessory nodes as a cancer therapeutic.

A couple of minor comments:

- I'm unsure of the point the authors are trying to make with the following statement, especially in terms of inhibition vs. attenuation.

"With accessory proteins being of immense relevance for the whole signalling machinery and operating particularly from the sideline, we propose that functional interference with a defined site of accessory proteins may attenuate rather than inhibit the signalling of hyperactivated RTK-RAS-MAPK axis".

- Also, the article could be significantly improved by a more expanded section of the future challenges of this type of approach.

Manuscript number:
COMMSBIO-21-0384-T

Title:
Accessory proteins of the RAS-MAPK pathway: Moving from the sideline to the front line

Authors:
Silke Pudewell, Christoph Wittich, Neda S. Kazemineh, Farhad Bazgir, Mohammad R. Ahmadian

Editor:
Dr. Karli Montague-Cardoso

Dear Editor and Reviewers:

Thank you very much for your consideration of our manuscript "Accessory proteins of the RAS-MAPK pathway: Moving from the sideline to the front line". We greatly appreciate your comments and open questions. We have responded them point by point below.

Authors' response to the reviewers' comments:

Reviewer #1:

Title; Review of Accessory proteins of the RAS-MAPK pathway: Moving from the sideline to the front line Paper summarizes the field of liquid-liquid phase separation in respect to RAS-RAF-MEK-ERK pathway. Previously, authors did discover the Galectin interaction with Ras and have written couple of reviews.

Any insights and implications? What are the major claims of the paper? Are they novel and will they be of interest to others in the community and the wider field?

Manuscript does provide novel insights into signaling cascade activation and highlights importance of scaffolding function of proteins on forming liquid-liquid phase separation.

Issues? Is the work convincing, and if not, what further evidence would be required to strengthen the conclusions? On a more subjective note, do you feel that the paper will influence thinking in the field?

Authors made a convincing point on importance of accessory proteins on signal activation and control of localization signal within the cells and cellular compartments. Whether paper will influence the thinking in the field is heavily depends on upcoming research articles and advancements in the field. I believe that scaffold proteins are important but field is still trying to distinguish between allosteric activations versus increased proximity for activation as it was the case for KSR protein.

1. Authors explain and exemplified in depth the accessory proteins that impact on signaling cascade activation. Could authors elaborate potential of these accessory proteins as synthetic lethal pairs in cancers as a part of "accessory proteins as therapeutic targets" section?

Authors' response: The phenomenon of synthetic lethality is a very interesting topic that we have not considered before. It is well possible that some accessory proteins are part of synthetic lethality pairs, and thus crucial for, e.g., oncogenic RAS signalling. SHOC2 has been identified as a potential SL pair with mutated KRAS (Aquirre and Hahn, 2018). However, cell type specificity and other mutations have to be taken into consideration and synthetic lethality has remained unproven as an approach to find effective new ways of tackling RAS mutant cancers. SHOC2 has been suggested as a potential SL pair with KRAS. However, cell type specificity and other mutations have to be taken into consideration.

Hence, it would be very interesting to conduct database analysis on accessory proteins in the context of synthetic lethal pairs in cancer.

2. Technological limitations for studying impact of each accessory protein in signaling cascade could be mentioned. While these proteins are critical, studying their functional nuances require better assays on pathway activation and spatiotemporal movement of individual proteins in the pathway.

Authors' response: This is a very interesting suggestion, which has been indeed insufficiently delineated in the *Concluding remarks* section. We have now extended this part as following:

Even though interest in accessory proteins has grown in the past few years, the possibilities to practically visualize them, track their pathway, and experimentally and selectively affect their functions in human cells are keys to address questions about their cell type specificities, subcellular distribution, and physical interactions in a context-dependent manner. To investigate the impact of an accessory protein in the context of RAS-MAPK signalling, we suggest the following approach: (i) It is necessary to first determine a cell line that expresses the gene related to the accessory protein of interest using quantitative real-time PCR. (ii) It is crucial to investigate the accessory protein at the endogenous levels. The overexpression studies cause in spite of their experimental advantages (Prelich, 2012) various difficulties. A prominent example is KSR overexpression that has been erroneously identified as a suppressor of RAS signalling. (iii) The major challenges faced and likely to be faced in near future are the difficult task of the direct use of antibodies post-purchase without careful validation (Acharya et al. 2017). It is of major importance to validate the antibody specificity by immunoblotting purified protein or protein fragments, and cell lysates overexpressing gene or gene fragments encoding the accessory protein. (iv) Cell fractionation and confocal imaging under stimulated and non-stimulated conditions will prove if the proteins pre-assemble in complexes with their binding partners (as predicted for KSR-MEK) and where they are located within the cell; as we expect the accessory proteins to orchestrate the RTK-RAS-MAPK signalling in specific subcellular compartments (e.g., plasma membrane, early endosomes, lysosomes, Golgi or ER). (v) Gene knockout cell lines, generated by CRISPR/Cas9 technology, will allow to measure the impact of the accessory proteins as positive or negative modulators of the RAS-MAPK pathway, by determining the p-ERK/ERK ratio. Moreover, this approach will give an idea about possible feedback or compensation mechanisms of accessory proteins among each other. Thus, exploring these concepts in greater detail will provide the framework for the future research that will fill existing gaps in our knowledge and expand our understanding of more effective therapies.

Following the same comment,

A. How strength of each accessory proteins varies and how researchers could identify novel accessory proteins? How to emphasize the temporary interactions on signaling pathway and how does feedback loops fit into stability of the activation clusters?

Authors' response: These are remarkable questions, which wait to be addressed. Future in-depth *in silico* studies will focus on sequence alignment and better understanding of kinetics and thermodynamic parameters of these mostly low-affinity and transient protein-protein interactions and the corresponding interaction network profile of various accessory proteins.

Additionally, anticipating studies aim to identify conserved regions among the accessory proteins by mutual associating protein assemblies, and shedding light on better understanding of how some accessory proteins bind to several pathway components simultaneously. Moreover, the accessory proteins will additionally influence temporary protein-protein interactions in the course of cellular phase separation process by increasing the dwell time of protein interactions and concentrations in one specific site of the cell.

Feedback loops, which may be mediated by posttranslational modifications, may perfectly fit in this concept, as accessory proteins may provide building blocks for signal protein complex assembly. Furthermore, accessory proteins may hold the signalling partners in close proximity, which allows a rapid interaction upon activation of upstream signalling.

B. Authors mentions array of proteins that functions as scaffold protein in RAS/RAF/MEK/ERK pathway as shown in figure 1. How many of these scaffold proteins needs to be present for enhanced signaling at a given time? Could cell compensate for inhibition of specific protein by increased expression of another scaffold protein?

Authors' response: This is a crucial question, which cannot be answered yet. It could be reasonable that attenuation of one pathway by functionally interfering with an accessory protein might shift the equilibrium towards another pathway. However, some of the accessory proteins are cell-type specific or act in a different upstream context. For example, Paxillin and KSR both bind RAF/MEK/ERK, the substitution of KSR by Paxillin would be questionable due to their context-dependent subcellular specificity of for example Paxillin at focal adhesions. Other proteins, for example GRB2 and SHP2, show quite well how fine-tuned the orchestration of receptor specific activation of downstream mechanisms is. The knock-out of individual accessory proteins would give a more explicit picture of the compensatory mechanisms and the possible shift towards other downstream pathways.

Overall, manuscript decently summarizes the field. With minor edit, it could be accepted. We would also be grateful if you could comment on the appropriateness and validity of any statistical analysis, as well the ability of a researcher to reproduce the work, given the level of detail provided.

Authors' response: This article is a compilation of published studies and does not have statistical analysis that should be reproduced.

Reviewer #2:

This is a well-researched review article that focuses on the functions of accessory proteins that control specific nodes of the RTK-RAS-MAPK signalling pathway. Accessory proteins are explained in terms of their function and are divided into i) scaffold, ii) adaptor, iii) anchoring, and iv) docking proteins, followed by highlights of well-studied domains, and the role they play in interfacing and controlling pathway dynamics. The second half of the article focuses on how dysfunction of RAS-MAPK accessory proteins can lead to diseases such as cancer. The major claim of the article is that RTK-RAS-MAPK accessory proteins should be a major focus of experimental cancer therapeutics going forward. The article is persuasive and makes a strong case for targeting accessory nodes as a cancer therapeutic.

A couple of minor comments:

- I'm unsure of the point the authors are trying to make with the following statement, especially in terms of inhibition vs. attenuation. "With accessory proteins being of immense relevance for the whole signalling machinery and operating particularly from the sideline, we propose that functional interference with a defined site of accessory proteins may attenuate rather than inhibit the signalling of hyperactivated RTK-RAS-MAPK axis".

Authors' response: Most classical approaches in cancer therapy abolish the RTK-RAS-MAPK pathway by inhibiting one or more signalling molecules. Direct inhibition of proteins alters the gene expression program and triggers feedback mechanisms in order to compensate for the loss of signal transduction, which may be the basis for drug resistance. Direct inhibition is toxic for healthy cells. In contrast, targeting accessory proteins does not directly affect the activity of the constituent members of signalling pathways and thus,

attenuates the signal flow in most cases but may not abolish it. This is most likely well tolerated for normal healthy cells but may regulate a hyperactivated diseased cell down to physiological levels.

- Also, the article could be significantly improved by a more expanded section of the future challenges of this type of approach.

Authors' response: Thank you for your positive response and your comments. We included a paragraph dealing with future challenges in the answer of question 2 from Reviewer #1. We will add this section to extend the *Concluding Remarks* part of the paper.

REVIEWERS' COMMENTS:

Reviewer #1 (Remarks to the Author):

No further requests. Thanks for writing this comprehensive review.

Reviewer #2 (Remarks to the Author):

The authors have appropriately addressed all comments. I support publication of this review.